# Usefulness of Contrast-Enhanced CT in a Patient with Acute Phlegmonous Esophagitis: A Case Report and Literature Review

**DOI:** 10.3390/medicina58070864

**Published:** 2022-06-28

**Authors:** Su Min Yun, Yeon Joo Jeong, Minhee Hwang, Geewon Lee, Ji Won Lee, Gwang Ha Kim, Jeong A Yeom

**Affiliations:** 1Department of Radiology, Pusan National University Hospital, Biomedical Research Institute, Pusan National University School of Medicine, Busan 49241, Korea; ppappaking94@gmail.com (S.M.Y.); lunar9052@hanmail.net (Y.J.J.); hmh8807@naver.com (M.H.); rabkingdom@naver.com (G.L.); monophobia00@hanmail.net (J.W.L.); 2Department of Internal Medicine, Pusan National University School of Medicine, Busan 49241, Korea; doc0224@chol.com; 3Department of Radiology, Pusan National University Yangsan Hospital, Yangsan-si 50612, Korea

**Keywords:** phlegmonous esophagitis, endoscopy, endoscopic ultrasound, computed tomography

## Abstract

Acute phlegmonous esophagitis is a very rare, life-threatening form of esophagitis, characterized by diffuse bacterial infection and pus formation within the submucosal and muscularis layers of the esophagus. We describe a case in which contrast-enhanced chest CT was useful for evaluating the severity of phlegmonous esophagitis, which was overlooked and underestimated by endoscopy.

## 1. Introduction

A phlegmon is a localized area of acute inflammation of soft tissues, and the term ‘phlegmon’ is used to describe inflammation related to a bacterial infection. Phlegmon can be associated with the formation of suppurative exudate or pus [1,2], and phlegmonous infections may involve the gastrointestinal tract. Acute phlegmonous esophagitis is a rare and life-threatening disorder, characterized by bacterial infection of the submucosal and muscularis layers of the esophagus [1,2,3,4,5,6,7,8]. We describe a case in which contrast-enhanced chest computed tomography (CT) was useful for evaluating the severity of phlegmonous esophagitis, which was overlooked and underestimated by endoscopy.

## 2. Detailed Case Description

A 76-year-old woman with no specific medical history visited an otolaryngology clinic because of neck pain and a foreign body sensation in her throat after eating Korean pancakes containing mussels. A laryngoscopic examination revealed no visible abnormalities and the patient returned home. However, the patient subsequently visited the emergency room due to worsening neck pain and fever development.

A laboratory examination revealed elevated C-reactive protein (26.72 mg/dL) with no evidence of leukocytosis (WBC count 9710 cell/μL, seg. neutrophils 73.1%). The initial anteroposterior chest radiography revealed mediastinal widening and patchy consolidation in both lower lobes (not shown). The patient then underwent contrast-enhanced chest CT for further evaluation. The CT images showed diffuse circumferential esophageal wall thickening with low-density submucosa along the whole esophagus, increased mediastinal fat attenuation, and peri-esophageal fluid collection (Figure 1). Esophagitis and mediastinitis were highly suspected, but there was no evidence of esophageal perforation. CT also depicted peri-bronchial ground-glass opacity and consolidations in both lower lobes (Figure 2), suggestive of bronchopneumonia. The sputum culture revealed the presence of *Streptococcus Viridans* groups. Endoscopy was performed to detect possible esophageal perforation, but no mucosal lesions were observed. Antibiotic (meropenem plus levofloxacin) treatment was started immediately to treat pneumonia and esophagitis. After 5 days of antibiotic treatment, the pneumonia improved, as determined by chest radiography. However, posterior mediastinal lesions obscuring both paravertebral stripes were observed (Figure 3), and, thus, follow-up contrast-enhanced chest CT was performed. The follow-up contrast-enhanced CT images (Figure 4A–C) revealed diffuse circumferential esophageal wall thickening with esophageal wall rim enhancement and an internal hypodense lesion, which was also hypodense when examined by pre-contrast-enhanced CT (Figure 4D). In addition, a hypodense linear tract was observed running from the esophageal lumen to the upper esophageal wall at the first thoracic vertebral level (Figure 5). Based on considerations of CT findings, acute phlegmonous esophagitis with partial tear of the esophageal wall was highly suspected.

For the diagnosis and management of acute phlegmonous esophagitis, the patient was transferred to our hospital, and follow-up endoscopy and subsequent endoscopic ultrasonography were performed. Esophageal mucosa had a normal appearance when examined by endoscopy, but a small opening was seen at the upper esophagus (Figure 6). EUS revealed a diffuse hypoechoic lesion at the submucosal and inner muscularis layers, extending 18 to 30 cm from the incisor (Figure 7), which was consistent with intramural abscess formation. Transendoscopic abscess drainage was performed using a hook knife. The procedure was terminated after confirming that there was no more drainage or bleeding. Pus culturing from abscess drainage was not performed. The patient was hospitalized and antibiotic (ciprofloxacin for 7 days) treatment was continued. Follow-up endoscopy was performed 5 days later, and revealed that the esophageal abscess was much improved, but esophageal stenosis was present around the abscess site. The patient was discharged after 11 days of hospitalization due to symptom improvement and normal laboratory results (CRP 1.83 mg/dL).

## 3. Discussion

This is the first report that we are aware of to be issued on the usefulness of contrast-enhanced chest CT for the evaluation of the severity of acute phlegmonous esophagitis caused by mucosal injury by a foreign body. The term ‘phlegmonous’ refers to pus formation by a diffuse inflammatory process, and phlegmonous infections can spread throughout the gastrointestinal tract. The stomach is the most commonly involved site. More than 100 such cases have been described in the literature, with a reported mortality rate of 42% [1,2,3]. Acute phlegmonous esophagitis is even rarer than phlegmonous gastritis, though a handful of cases of phlegmonous esophagitis with or without stomach involvement have been reported [1]. In our case, phlegmonous changes were confined to the esophagus.

The pathogenesis of acute phlegmonous esophagitis is unclear. The reported predisposing factors include immunosuppression, alcoholism, diabetes mellitus, old age, and mucosal injury [2]. Its pathogenesis involves damage to the intestinal tract, resulting in susceptibility to bacterial infections [1]. In our case, a mussel shell fragment was probably responsible for injuring the esophageal wall. This is the first reported case of phlegmonous esophagitis attributed to an esophageal wound caused by food [7,8]. Phlegmonous infections usually involve the submucosal layer of the gastrointestinal tract, and, thus, endoscopic diagnosis is difficult and often delayed. 

A CT examination and EUS can provide an accurate, prompt diagnosis (Figure 8). CT imaging findings after intravenous contrast administration are specific for acute phlegmonous esophagitis, as the images depict an intramural circumferential hypodense esophageal area surrounded by peripheral rim enhancement [1,2,3,4,5,6], which are typical findings for phlegmonous esophagitis. Sometimes air bubbles are present within a thickened esophageal wall, indicating the presence of infection caused by gas-forming pathogens and a diagnosis of phlegmonous esophagitis [2,3,4,5,6]. The radiologic differential diagnosis of diffuse esophageal wall thickening includes dissecting intramural hematoma, tubular duplication of the esophagus, or corrosive or reflux esophagitis [1,2]. Dissecting intramural hematoma is an uncommon form of esophageal injury and is characterized by bleeding contained between the mucosa and the muscular layers of the esophagus. It is different from the mucosal tear seen in Mallory-Weiss syndrome and from the transmural rupture in Boerhaave’s syndrome. Dissecting intramural hematoma may occur within the esophageal submucosal layer following the dissection of mucosa. In cases of dissecting intramural hematoma, the esophageal wall is visualized as a hyperdense lesion on non-enhanced CT images [5]. There are two types of esophageal duplication, namely, cystic and tubular, and the cystic type is more common. Approximately 40% of esophageal duplications are found in the upper or middle third of the esophagus [9]. On CT, tubular esophageal duplication is observed as a well-defined mass with internal fluid density and is sometimes associated with thin enhancement due to mucosal lining. However, the majority of tubular esophageal duplications do not exhibit enhancement. Unlike cystic duplications, tubular duplications usually communicate with a normal esophagus [10]. In most cases, tubular esophageal duplication and phlegmonous esophagitis have similar CT findings [5]. Patients with tubular duplication of the esophagus are likely to have no symptoms or signs [2]. On the other hand, corrosive esophagitis has CT findings dependent on its severity. In cases of severe esophagitis, regardless of its etiology, the CT findings include diffuse esophageal thickening, submucosal edema, and mucosal enhancement, and depend on the stage of the disease (acute < 10 days, subacute 10–20 days, or chronic > 21 days) [11,12]. Corrosive esophagitis usually affects the middle or lower third of the esophagus and its CT findings are nonspecific. However, patients with corrosive esophagitis invariably have a history of acid or base ingestion, which is crucial for diagnosis [1]. Reflux esophagitis is often accompanied by a sliding esophageal hernia and usually involves the mid-to-lower esophagus without peripheral rim enhancement on CT images [1]. EUS findings are diffuse thickening with hypoechoic lesions in the esophageal submucosal layer, and in cases of reflux esophagitis, EUS enables detailed visualization of abscesses in the submucosal layer. However, CT is a better option for evaluating the overall lesion extent and is more useful for diagnosing accompanying complications, such as esophageal perforation, mediastinitis, and pleural effusion, and determining additional surgical requirements.

The treatment options for phlegmonous esophagitis include managing the infection with antibiotics, internal pus drainage, or surgical intervention [4]. There is no standardized treatment because of the paucity of cases [3]. Inflammation and infection management is important to prevent the onset of a septic condition or complications such as esophageal necrosis, esophageal stricture progression, perforation, mediastinitis, or peritonitis. Adequate nutrition is also critical for long-term medical management [3,4].

## 4. Conclusions

Phlegmonous esophagitis is easily overlooked because it is a rare disease with no specific signs or symptoms. Therefore, radiologists should be well aware of the CT findings and risk factors of this disease, and advise clinicians on its appropriate management.

We report a rare case of acute phlegmonous esophagitis in which contrast-enhanced CT was useful for diagnosis and evaluating extent, which was underestimated by initial endoscopy. This section is mandatory and should contain the main conclusions regarding the research.

## Figures and Tables

**Figure 1 medicina-58-00864-f001:**
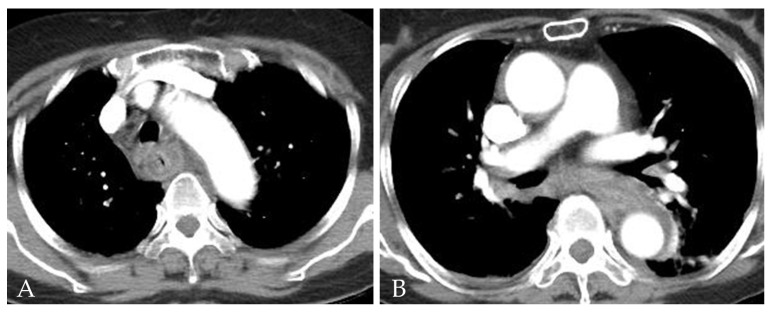
A 76-year-old woman with acute phlegmonous esophagitis. (**A**,**B**) Axial contrast-enhanced CT images show diffuse circumferential esophageal wall thickening with low-density submucosa along the entire esophagus, increased mediastinal fat strand, and peri-esophageal fluid collection.

**Figure 2 medicina-58-00864-f002:**
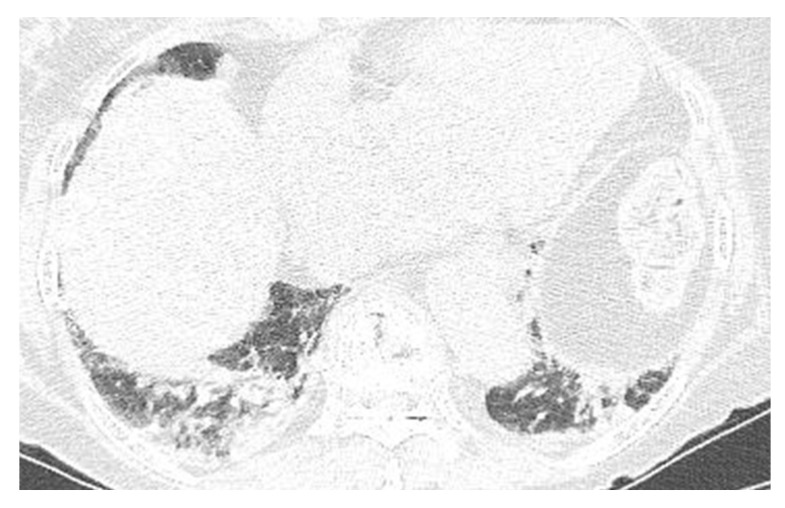
A 76-year-old woman with acute phlegmonous esophagitis. Lung window of an axial CT image at the liver dome level shows peri-bronchial ground-glass opacities and consolidations in both lower lobes.

**Figure 3 medicina-58-00864-f003:**
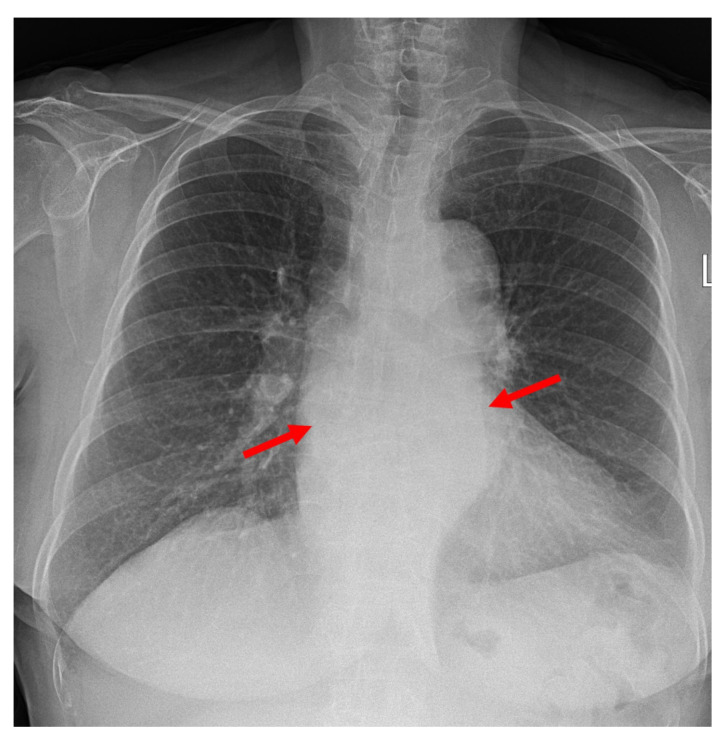
A 76-year-old woman with acute phlegmonous esophagitis. Chest posteroanterior radiograph obtained 5 days after the initial CT image shows mass-like opacity (arrows) that obscured both paravertebral stripes, suggestive of a posterior or paravertebral mass lesion.

**Figure 4 medicina-58-00864-f004:**
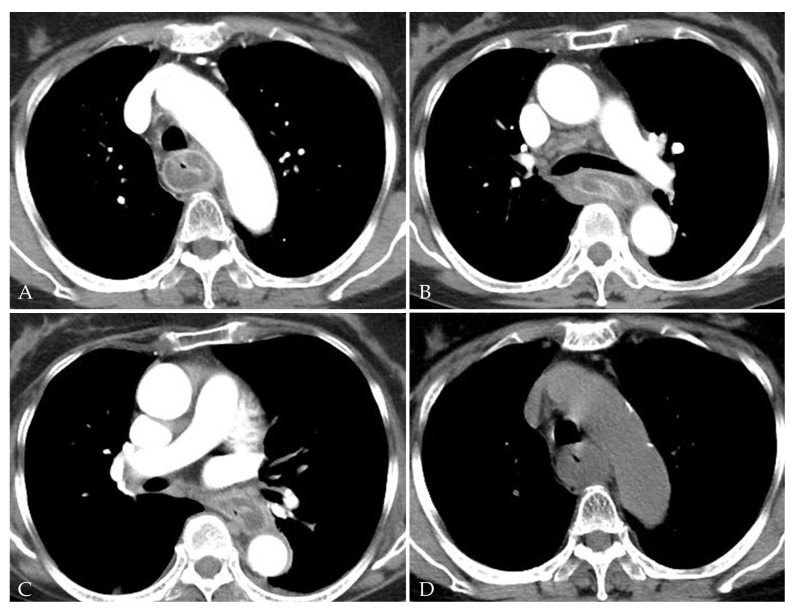
A 76-year-old woman with acute phlegmonous esophagitis. Axial contrast-enhanced CT images (**A**–**C**) show diffuse circumferential esophageal wall thickening with esophageal wall rim enhancement and an internal hypodense lesion, which was hypodense on pre-contrast-enhanced CT image (**D**).

**Figure 5 medicina-58-00864-f005:**
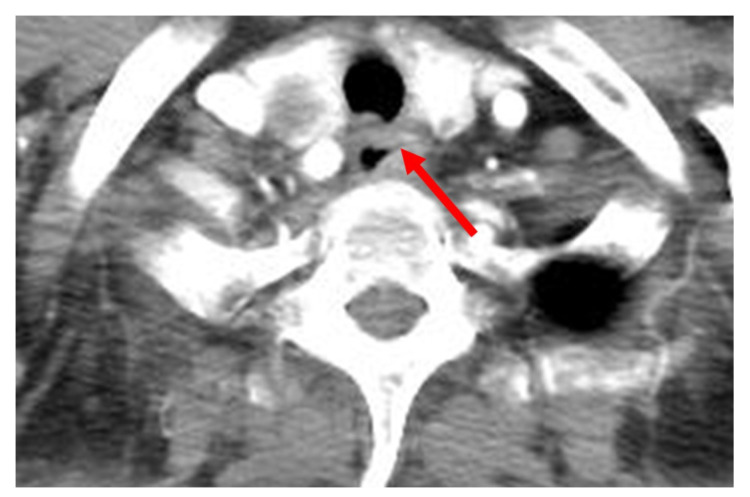
A 76-year-old woman with acute phlegmonous esophagitis. Axial contrast-enhanced CT image at 1st thoracic vertebral level shows a hypodense linear tract (arrow) running from esophageal lumen to upper esophageal wall, suggestive of partial tear of the esophageal wall.

**Figure 6 medicina-58-00864-f006:**
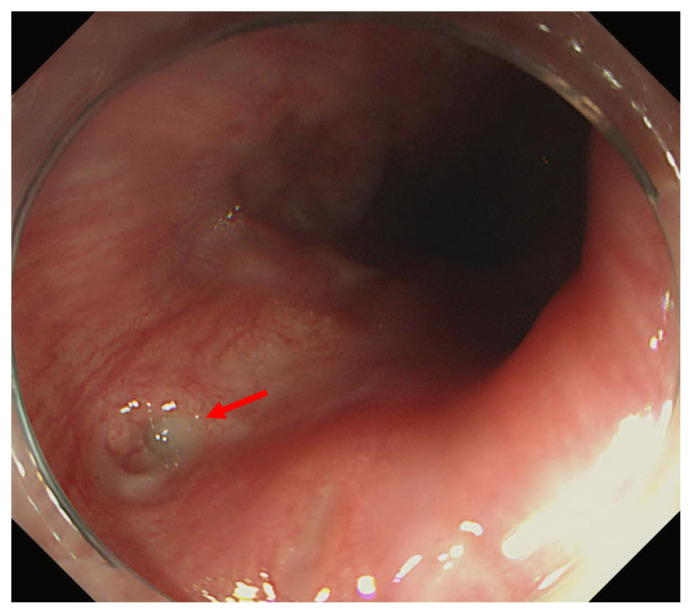
A 76-year-old woman with acute phlegmonous esophagitis. Endoscopy shows normal esophageal mucosa and a small opening (arrow) at the upper esophagus.

**Figure 7 medicina-58-00864-f007:**
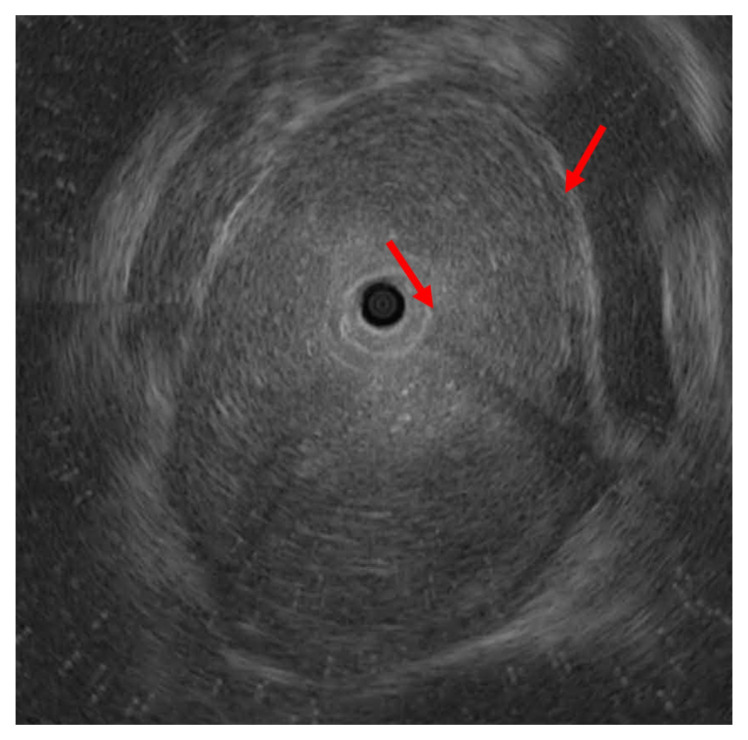
A 76-year-old woman with acute phlegmonous esophagitis. Endoscopic ultrasonographic image shows diffuse hypoechoic lesion (arrows) at the submucosal and inner muscularis layers of the esophagus.

**Figure 8 medicina-58-00864-f008:**
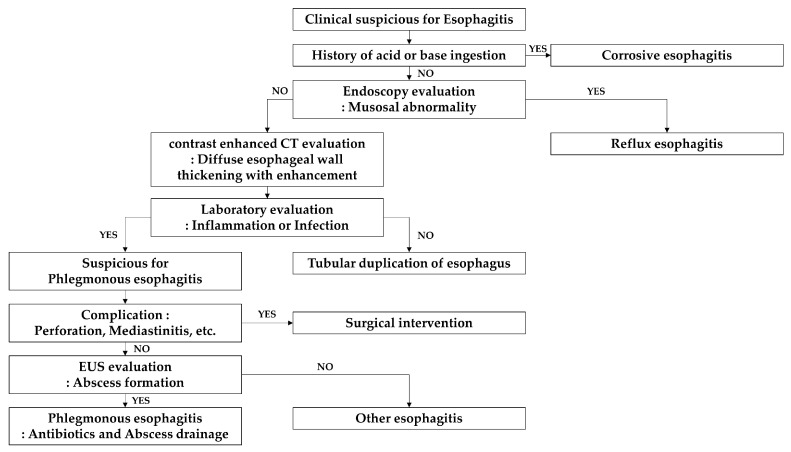
Flow chart of diagnosis of esophagitis with CT examination and EUS.

## Data Availability

Not applicable.

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
