# Peer review of "Usefulness of Contrast-Enhanced CT in a Patient with Acute Phlegmonous Esophagitis: A Case Report and Literature Review"

_medicina, 2022, doi:10.3390/medicina58070864_

Round 1

Reviewer 1 Report

This is a case report of a rare condition and therefore worthwhile publishing. Minor changes suggested include the avoidance of using abbreviations.

1. One example of this is page 2 line 45. What is peri-bronchial GGO?

2. Page 5 line 105: I would change to "this is the first report we are aware off to be issued on the usefulness...." It is hard to be sure that you're reporting on the first ever case of anything......

3. Minor English errors: Please go over carefully.

Author Response

This is a case report of a rare condition and therefore worthwhile publishing. Minor changes suggested include the avoidance of using abbreviations.

-> Thank you for your comments.

* R1-1. One example of this is page 2 line 45. What is peri-bronchial GGO?

-> As you pointed out, we revised GGO as ground glass opacity.

* R1-2. Page 5 line 105: I would change to "this is the first report we are aware off to be issued on the usefulness...." It is hard to be sure that you're reporting on the first ever case of anything......

-> Thank you for pointing this out. We revised the sentence as you recommended.

* R1-3. Minor English errors: Please go over carefully.

->  Our manuscript has already been serviced by a professional English proofreading company, NURISCO. A certificate will be submitted along with the revised document. As you pointed out, we checked carefully if English editing was necessary once again.

Reviewer 2 Report

Congratulation to the medical team and authors for successfully treating such a rare case with a very high risk of mortality. The case was well described and the images are clear and very informative. 

Can the authors please tell us what specific antibiotics are used for this case, and for how long? Did the pus from the oesophageal abscess grow any specific organisms? This is highly important given the paucity of literature around this condition. 

Minor correction: legend for GGO abbreviation in line 45 should be provided in full first (Ground Glass Opacification).

Author Response

Congratulation to the medical team and authors for successfully treating such a rare case with a very high risk of mortality. The case was well described and the images are clear and very informative.

-> Thank you for your comments.

* R2-1. Can the authors please tell us what specific antibiotics are used for this case, and for how long? Did the pus from the oesophageal abscess grow any specific organisms? This is highly important given the paucity of literature around this condition.

-> Thank you for pointing this out. We reported types and duration of antibiotics used as you recommended. Antibiotics (Meropenem plus levofloxacin) treatment was started immediately to treat pneumonia and esophagitis. And the patient was hospitalized, and antibiotic (ciprofloxacin for 7 days) treatment was continued.

Pus culture from abscess drainage was not performed in our patient. This is because the clinician judged that culture was meaningless because antibiotic treatment had been done for about a week before abscess drainage.

* R2-2. Minor correction: legend for GGO abbreviation in line 45 should be provided in full first (Ground Glass Opacification).The study is well performed and well written with clinical relevant observations.

-> As you pointed out, we revised GGO as ground glass opacity.
